# Effect of Copper Antifouling Paint on Marine Degradation of Polypropylene: Uneven Distribution of Microdebris between Nagasaki Port and Goto Island, Japan

**DOI:** 10.3390/molecules29051173

**Published:** 2024-03-06

**Authors:** Hisayuki Nakatani, Kaito Yamashiro, Taishi Uchiyama, Suguru Motokucho, Anh Thi Ngoc Dao, Hee-Jin Kim, Mitsuharu Yagi, Yusaku Kyozuka

**Affiliations:** 1Polymeri Materials Laboratory, Chemistry and Materials Engineering Program, Nagasaki University, 1-14 Bunkyo-machi, Nagasaki 852-8521, Japan; bb52123643@ms.nagasaki-u.ac.jp (K.Y.); bb52122606@ms.nagasaki-u.ac.jp (T.U.); motoku@nagasaki-u.ac.jp (S.M.); anh.dao@nagasaki-u.ac.jp (A.T.N.D.); 2Organization for Marine Science and Technology, Nagasaki University, 1-14 Bunkyo-machi, Nagasaki 852-8521, Japan; kyozuka@nagasaki-u.ac.jp; 3Graduate School of Fisheries and Environmental Sciences, Nagasaki University, 1-14 Bunkyo-machi, Nagasaki 852-8521, Japan; heejin@nagasaki-u.ac.jp (H.-J.K.); yagi-m@nagasaki-u.ac.jp (M.Y.)

**Keywords:** microplastics, Cu-based paint, polypropylene, enhanced degradation

## Abstract

Microplastics (MP) encompass not only plastic products but also paint particles. Marine microdebris, including MP, was retrieved from five sampling stations spanning Nagasaki-Goto island and was classified into six types, primarily consisting of MP (A), Si-based (B), and Cu-based (C) paint particles. Type-A particles, i.e., MP, were exceedingly small, with 74% of them having a long diameter of 25 µm or less. The vertical distribution of type C, containing cuprous oxide, exhibited no depth dependence, with its dominant size being less than 7 μm. It was considered that the presence of type C was associated with a natural phenomenon of MP loss. To clarify this, polypropylene (PP) samples containing cuprous oxide were prepared, and their accelerated degradation behavior was studied using a novel enhanced degradation method employing a sulfate ion radical as an initiator. Infrared spectroscopy revealed the formation of a copper soap compound in seawater. Scanning electron microscopy/energy-dispersive X-ray spectroscopy analysis indicated that the chemical reactions between Cl^−^ and cuprous oxide produced Cu^+^ ions. The acceleration of degradation induced by the copper soap formed was studied through the changes in the number of PP chain scissions, revealing that the presence of type-C accelerated MP degradation.

## 1. Introduction

When plastic litter enters the sea, it disperses and causes microplastic (MP) pollution [1,2,3,4,5,6,7,8,9]. MPs are small plastic particles of less 5 mm in diameter with various shapes [10,11]. Their composition depends on industrial plastic production, primarily consisting of fragments of polypropylene (PP), polyethylene (PE), and polystyrene (PS) resulting from degradation caused by sunlight exposure [12,13,14]. However, the natural rate of MP production due to sunlight is very slow; thus, accelerated MP production methods are needed to experimentally confirm the impacts of MP exposure on ecosystems. In our previous study [15], we conducted PP degradation in seawater using an advanced oxidation process (AOP) with a sulfate ion radical (SO_4_•^−^) as a highly efficient initiator for plastic degradation. The combination of seawater and the SO_4_•^−^ initiator resulted in the excellent acceleration of the degradation process under pH control. It is noted that marine MP formation involves autoxidation in the presence of various factors, such as paint particles, in addition to seawater. Very little research has been conducted on microplastics and their interactions with heavy metal paint fragments, i.e., the surface chemistry of microplastics. For instance, antifouling paint is routinely applied to submerged structures in boats to prevent colonization by various organisms such as algae and invertebrates [16,17,18,19]. Common antifouling formulations employ an active metal oxide pigment and various organic or organo-metallic booster biocides embedded in, or linked to, a polymeric matrix. In particular, cuprous oxide (Cu_2_O) is the typical choice of pigment along with Cu(I) thiocyanate and Zn(II) oxide as co-additives. Thus, antifouling paint often contains cuprous oxide. In this study, we retrieved marine tiny particles from surface water and a water column (two vertical samples collected at five sampling stations) in the East China Sea near Nagasaki City, Nagasaki Prefecture, Japan, using a boat to confirm the presence of paint particles containing cuprous oxide or Si compounds. Nagasaki Port is a port for small high-speed vessels (fishing boats), while the sampling station on the Goto side is located on the route of large vessels (ferries). The small vessels use silicon-based paint, and the large vessels use copper-based paint. Therefore, these five sampling stations are expected to have different paint fragment distributions. The retrieved samples were examined by scanning electron microscopy/energy-dispersive X-ray spectroscopy (SEM/EDX) analysis to confirm the presence of paint particles containing cuprous oxide. In addition, the difference between the ratio of MP and paint particles at each sampling site was investigated. From these results, we studied the involvement of the presence of paint particles containing cuprous oxide in the natural phenomenon of MP disappearance in the sea within a few years [20]. Moreover, to investigate the effect of cuprous oxide on the degradation rate of MP in seawater in detail, a novel ”enhanced degradation method” was developed with polymer samples containing cuprous oxide using a sulfate ion radical as an initiator.

## 2. Results and Discussion

### 2.1. Types of Marine Tiny Particles and Their Distribution Behavior

In this study, the sizes, shapes and elemental composition ratios of particles retrieved from the sea were determined using SEM/EDX analysis. Through SEM/EDX analysis, the retrieved marine tiny particles were classified into six types: A: MP; B: Si-based paint particles; C: Cu-based paint particles; D: Other paint particles; E: Al coating laminate film debris; F: Shell pieces. Here, the classification was based on elemental analysis using EDX while taking into account the shape. Type A was assigned to particles composed predominantly of C content and to those with less 1 mol% Si and/or Cu contents to account for errors in the elemental composition analysis by EDX [21]. The proportion of materials used in disposal plastics is more representative because the proportion of secondary MPs formed from non-disposable plastics is not high, except for tire rubber. PE, PP and PS are produced globally on an enormous scale. In particular, PP and PE represent 22% and 23% of Japanese resin manufacturing (in 2018), respectively. PS has also been used in the fishing industry as Styrofoam. Consequently, type A mainly comprised PE, PP, and PS [15]. Types B and C were assigned a Si or Cu content of over 1 mol%, respectively. Type D was assigned particles having a high content of various elements including carbon, and type E was designated for those having a film-like shape and high contents of Al and C. Type F was assigned to shell fragments, which were ceramic fragments in shape and had an elemental composition ratio of Ca:C:O≈1:1:3. In addition, the shapes of these types were different from plankton debris, making them distinguishable by appearance. Furthermore, differences existed in the constituent elements and composition ratios of elements between them. Plankton debris primarily comprised polysaccharides, such as cellulose, and the O/C molecular ratio was much higher (ca. 0.8). All results of the long diameter (d), EDX analysis, and materials of the retrieved samples are summarized in the Appendix A. Figure 1 and Appendix A show the SEM photographs and their EDX analysis of the three main types, respectively. The shapes of types A and B were diverse, while the shape of type C mostly consisted of an aggregate of fine particles, as depicted in Figure 1. The EDX analysis showed that the fine particles were composed of C, Cu, and Zn compounds. Type C is considered typical antifouling paint consisting of an organic paint component (polymers such as polyurethane, polyacrylic, and epoxy), copper additive, and zinc co-additive compounds [17,18]. It uses cuprous oxide as the main antifouling substance, and other metal compounds such as zinc–pyrithione (ZnPT) and copper–pyrithione (CuPT) are used as auxiliary antifouling substances [18,19]. The paint is used to coat ship bottoms to prevent the growth of fouling organisms, including algae, barnacles, and bivalves. Cuprous oxide remains as fine particles in seawater for a long time due to its poor water solubility and could have implications for the autoxidation of type-A MPs, such as PP, PS, and PE. Chan et al. reported that cuprous oxide reacted with degraded PE compounds and produced copper carboxylate salts, exhibiting highly catalytic performance for autoxidation at relatively lower temperatures [22]. Therefore, the presence of paint particles containing cuprous oxide could affect the amount of MP, but no research has confirmed this possibility. In fact, as type A becomes a carboxylic acid due to the progression of degradation (autoxidation), it gradually changes to a soluble ionic species. Type C, which contains cuprous oxide, serves as a source of copper ion species to produce the copper carboxylate, which could affect the quantity ratio of type-A MPs by causing more severe degradation in the sea. Figure 2a,b show the percentage of MP and microdebris at each sampling station. All the percentages of type A at the surface (B) were higher than those at the 50 m depth (D) at the same horizontal sampling (S1, S3, S5, S7 and S9) spots (see Appendix A). The easy floating behavior on the sea surface reflects that type A was mainly composed of PE, PP, and expanded PS (EPS). In contrast to type A, type B (Si-based paint particles) showed that the percentages in the 50 m depth were higher than those on the surface at all horizontal sampling spots. Type B is a type of fouling-release coating and consists of cross-linked siloxane structures such as silicone rubber [23]. The relatively high specific gravity of the fragment causes it to sink into the sea, possibly contributing to the higher percentage at the 50 m depth. Conversely, the percentages of type C showed no depth dependence, despite the presence of high-specific-gravity cuprous oxide. The relative abundance distribution of microdebris such as types B and C was not determined by depth dependence alone. The distribution of the microdebris abundance should be considered in terms of several parameters: size, density, feed rate, and degradation rate. As shown in Figure 2, type B was more abundant at the S1 spot near Nagasaki, and type C was more abundant at S7 and S9 spots near Goto Island. The bias in the presence ratio of type B and type C is closely related to the fact that the S1 spot is next to a fishing port (New Nagasaki Fishing Port) and in the path of high-speed vessels such as fishing boats, whereas S7 and S9 spots are in the path of ferries, which are large vessels. Types B and C paints are for fast and large vessels, respectively, and the bias in the presence ratio was due to the different types of vessels passing over the sampling spots. Thus, the time (and season) of sampling would have little effect on the presence-ratio bias. Figure 3 shows the relationship between the Cu content ratio and the long diameter of the type-C samples. The long diameters of most of the samples were very small, less than 7 μm. Cuprous oxide had a copper content of 67 mol%, and the copper content found in most samples was below this value. Although there is no clear correlation between the Cu content ratio and long diameter, i.e., fragmentation size, many fragments were less than 7 μm in size and contained 30–50 mol% of Cu content. This irrelevance between the size and Cu content was due to the heterogeneous structure of cuprous oxide-based paint. In addition, the copper-containing particles could also be copper pyrithione or copper thiocyanate. However, these copper compounds are employed as co-additives in cuprous oxide antifouling paint, and the content is much lower. The copper component in the particles observed by SEM/EDX observation appeared to be almost exclusively cuprous oxide. Appendix A shows that the long diameters of most of the type-B samples were less than 15 μm and were considerably larger than those of type C. On the other hand, most of them contained less than 20 mol% of Si. This relatively narrow range of Si content suggests that type B was formed by a homogeneous structure, such as silicon rubber. Figure 4 shows the particle numbers of type C and their Cu content ratio at the surface and 50 m depth spots at all sampling stations. The predominant Cu content ratio of exfoliated particles was estimated to be 31–40 mol% in both spots, as shown in Figure 4. In addition, the surface spot samples showed a biased distribution behavior toward relatively low Cu content. It appears that the change in the dominant Cu content was small when the exfoliation occurred. The high Cu content at the time of exfoliation was maintained, and the type-C particle sank very slowly into the sea. Generally, cuprous oxide particles of 2–4 μm are used as raw materials for antifouling paint [24]. Thus, the aggregates of two or three cuprous oxide particles, fixed together using the paint resin binder, are considered preferentially exfoliated as type C from the predominant copper content (31–40 mol%) and particle size (<7 μm). Considering Stokes’ law, it seems that the type-C particle has a very slow sedimentation rate due to its small size. These results suggest that the particles with 31–40 mol% Cu content remain for a long time at a shallow depth of up to 50 m below sea level. Naturally, the cuprous oxide components gradually disappear from the paint matrix. This disappearance causes the paint particles to become lighter and to float back to the sea surface, resulting in a smaller difference in the abundance between the surface and 5 m depth sampling spots due to such fluctuations as equilibria. The size of type C is independent of the Cu content ratio because the loss of cuprous oxide occurs after the exfoliation process.

### 2.2. Ionization Behavior of Cuprous Oxide in Seawater and Its Effect on PP Degradation

Type C contains tiny cuprous oxide particles and could exhibit highly catalytic performance for autoxidation. In particular, the characteristics of fine particles and their long-term existence near the sea surface are thought to have implications for the autoxidation of floating plastic litters of PP, PE, and EPS to MPs. Figure 5 shows the percentage of long diameter MP (type A) samples retrieved from the sea. Of the MPs recovered, 74% (113 particles out of 153 particles) were very small, with a long diameter of 25 μm or less. Therefore, the surface area per unit volume is so large that the catalytic activity of copper should be strongly reflected. Such small MPs are expected to be degraded and be lost in a short time, even at low copper concentrations. However, it is very difficult to study the degradation behavior of small plastic samples in detail. The degradation in the plastic samples with a few cm diameter is desirable. On the other hand, cm-sized plastic samples take a long time to degrade in weathering tests. To clarify the relationship between the type C, i.e., cuprous oxide presence and MP loss, it is necessary to use a degradation accelerator system that can accelerate the degradation of such cm-sized plastic samples. Appendix A shows the comparison of the degradation behavior between normal weathering degradation and the novel enhanced degradation in seawater. Normal weathering degradation occurs only on the surface of target polyolefin samples such as PP. Since the cuprous oxide particles also have difficulty penetrating the sample matrix, their effect on the sample degradation is limited to the surface and is therefore negligible. Therefore, a typical xenon lamp weathering system requires a large amount of samples to study the effect of cuprous oxide deposition on MP degradation in the sea. In addition, because the rate of degradation in seawater is very slow [25], sampling must be conducted over a long period of time. To obtain the required amount for analysis, it is necessary to use a degradation method that accelerates the degradation rate while simultaneously degrading within the sample [25]. Therefore, we developed a novel “enhanced degradation method” using a sulfate ion radical in seawater to homogeneously degrade the entire sample containing cuprous oxide, as shown in Appendix A. The enhanced degradation method resulted in a sufficient amount of degraded sample for analysis in a short period by homogeneously degrading the entire sample at a higher degradation rate. Figure 6 shows the reaction path of metal soap production via PP autoxidation and the FT-IR spectrum of degraded PP containing 5 phr cuprous oxide using the enhanced degradation method in seawater for 3 days. The spectrum reveals that even in seawater, autooxidation led to the production of γ-lactone and carboxylic acid along the reaction path depicted in the scheme. In addition, a broad peak assigned to the metal soap compound can be observed between 1700 and 1600 cm^−1^ [22]. The Cu metal soap acts as a catalyst for autoxidation and as a surfactant, inducing the fragmentation of polyolefins such as PP in water [26]. A potential environmental impact can be expected from the synthesis of metal soap via autoxidation, which would accelerate the rate of MP formation in the sea. Appendix A shows the color change of PP containing a 5 phr cuprous oxide film induced by the enhanced degradation method. When the film was degraded in seawater, the color changed from reddish brown due to cuprous oxide to pale blue in just 3 days. This color change was due to the valence change of copper ions. We have previously reported that seawater drastically enhances PP degradation performance due to the presence of sulfate ion radicals under pH control [15,25]. Cl^−^ usually reacts with OH• generated by solar irradiation in seawater and yields Cl_2_. The resulting Cl_2_ then reacts with H_2_O and forms ClOH, with the equilibrium of the two reactions dependent on pH, as follows [27]:
Cl_2_ + 2H_2_O ⇄ ClOH + Cl^−^ + H_3_O^+^, ClOH + 2H_2_O ⇄ ClO^−^ + H_3_O^+^
(1)

Because the pH of seawater is around 8.2, the equilibrium favors the less reactive ClO^−^, thus suppressing PP degradation in seawater [27]. Therefore, we adjusted the pH to shift the equilibrium toward ClOH in Equation (1) [15]. In our previous work [15], SO_4_•^−^ was employed as an initiator in seawater, and Cl^−^ converted it to OH• during the PP degradation process. In addition, some parts of SO_4_•^−^ were gradually converted to SO_4_^2−^, which affected the pH of the alkaline seawater. As shown in the Experimental section, the enhanced degradation method was carried out for a predetermined number of days as one set/one day. The pH of the solution was changed from 8.2 to 3 and returned to 8.2 again during one set. The pH in this work varied in a complex manner, from a pH value of 8.2 in the marine environment to a lower value and back to 8.2 again. In addition, Cu^+^ reacted with ClOH and was converted to Cu^2+^ according to Equation (2) [28].
Cu^+^ + HOCl → Cu^2+^ + OH• + Cl^−^(2)

The solubilization, namely ionization, of cuprous oxide was accelerated by pH adjustment and the use of ClOH compared to the marine environment. The color change of the film reflected Cu^2+^ production, showing that OH• is generated as an initiator of autoxidation [see Equation (2)]. As shown in Appendix A, the degradation rate of the film in pure water was significantly slower than that in seawater, and the film finally turned white 45 days after the start of degradation. Appendix A shows the degradation time dependence of the FTIR spectra and carbonyl index values of the films in pure water and seawater using the enhanced degradation method. The peak at 640 cm^−1^ is attributed to cuprous oxide [23,29]. In the seawater degradation, the peak decreased rapidly after only 1 day and almost disappeared after 3 days. As shown in Appendix A, both cuprous oxide peaks rapidly disappeared within 1 day when the salinity concentrations were increased two and three times. Cuprous oxide rapidly dissolves and ionizes in seawater. The ionization behavior of cuprous oxide in seawater has been interpreted as follows:½Cu_2_O(s) + H^+^ + 2Cl^−^ ⇄ CuCl_2_^−^ + ½H_2_O(3)
CuCl_2_^−^ + Cl^−^ ⇄ CuCl_3_^2−^(4)
where Equation (3) is reversible but kinetically slower, and Equation (4) is reversible and faster [19,30]. As described in the Experimental section, seawater containing K_2_S_2_O_8_ was added after 12 h, and samples were collected daily to restart the degradation with fresh seawater containing K_2_S_2_O_8_. Conversely, cuprous oxide, which is mixed with PP, is not supplemented and is quickly eluted by ionization. It is difficult to provide direct evidence for the presence of CuCl_2_^−^ and CuCl_3_^2−^ in Equations (3) and (4) under the specific pH condition. Appendix A shows that the color change of the film from reddish brown to pale white occurs in seawater within only 3 days, suggesting that Cu^+^ (CuCl_2_^−^ and/or CuCl_3_^2−^) is rapidly oxidized to Cu^2+^ in the presence of dissolved oxygen. Appendix A shows the SEM/EDX analysis of the degraded PP containing 5 phr cuprous oxide by the enhanced degradation method in seawater for 3 days. The molar ratio (Cu/O) values of the identified particles are in the range of 0.35–1.38, significantly lower than those in cuprous oxide (Cu/O = 2). These low Cu/O values indicate that the chemical reactions of Equations (3) and (4) proceed to produce Cu^+^ ion species from cuprous oxide. Figure 7 shows the degradation-time dependence of FTIR spectra and CI values of various PP samples at different salinity concentrations of seawater, as well as SEM/EDX analysis of the degraded PP containing 5 phr cuprous oxide by the enhanced degradation method in seawater at normal salinity concentration for 6 days. In the case of PP degradation in seawater [15], a reaction occurs between Cl^−^ and OH• to produce Cl•; then, two Cl• atoms couple to produce Cl_2_, which then reacts with H_2_O to produce ClOH, with the equilibrium depending on pH. The pH value of K_2_S_2_O_8_ in the seawater solution (normal salinity concentration) decreased from ca. 8.2 to ca. 3 during the daily exchange (see Section 3) [15]. This procedure ensures a bias towards the ClOH-rich equilibrium for a period of time before replacement with a fresh K_2_S_2_O_8_ seawater solution. The ClOH has a longer lifespan [27] and migrates deep into the interior of PP before dissociating into radicals and initiating autoxidation [27], simultaneously degrading the interior and surface of PP. The ClOH-rich equilibrium depends on the pH of the solution, as mentioned above. Thus, the higher-salinity concentrations will not only affect the Cu^+^ species but also the ClOH production. As shown in Figure 6, all PP samples containing cuprous oxide have ca. 10–20% higher CI values after 6 days. As described above, the presence of Cu compounds was observed by the EDX measurements up to 3 days, but not at 6 days. This result suggests that the acceleration of PP degradation is not only caused by solid Cu compounds, such as cuprous oxide, but also by soluble compounds, such as Cu^+^, in small amounts. The broad IR peak (between 1700 and 1600 cm^−1^) assigned to the metal soap, i.e., copper carboxylate, was observed up to 15 days in each spectrum in Appendix A. The Cu^+^ adsorbed by ionic bonds as copper carboxylate is typically released due to equilibrium and generates OH• according to the reaction of Equation (2). In addition, copper hydroperoxide (COO^−^ Cu^+^) appears to exist as an adsorbed compound of Cu^+^. The broad peak around 3300 cm^−1^ can be observed in the 1-day sample at two times the salinity concentration, 3 and 6 days at three times the salinity concentration (see Appendix A). The peak would be assigned to copper hydroperoxides (COO^−^ Cu^+^ and/or COO^−^ Cu^2+^), which originate from the bonding between hydroperoxides in the degraded PP [31,32] and copper ion species. The main reason why the CI values for each sample vary significantly after the 6-day degradation time is that the autoxidation process continues and side reactions other than the autoxidation reaction begin. Because the side reactions that produce esters and carboxylic acids are influenced by various factors, such as pH values, the extent of progress of the side reactions appears to vary significantly with salinity concentration. Therefore, to evaluate the progress of autoxidation, which is the main degradation reaction influenced by copper ions, it is more appropriate to examine the degree of molecular weight reduction, namely, the increase in the number of molecular chain scissions. Table 1 and Figure 8 show the molecular weight information of various samples and their 1/Mn × 10^5^ plotted versus degradation time. The reciprocal of Mn (1/Mn) is the number of polymer chain ends, and an increase in this value implies an increase in the number of polymer chain ends, i.e., an increase in the number of polymer chain scissions. Both PP samples containing 5 phr cuprous oxide in the normal and three-times-salinity concentration of seawater samples have ca. 30–50% higher 1/Mn × 10^5^ values than samples without cuprous oxide after up to 9 days. This result indicates that a copper ion, probably Cu^+^, promotes autoxidation. In addition, there is almost no difference in the value after 15 days among these samples. After 14 solution exchanges, the adsorbed copper ions are removed, and the amount of adsorption certainly decreases. The convergent behavior suggests that copper ion species promote autoxidation. The convergence of the difference in the number of polymer chain scissions supports the hypothesis that copper ion species promote autoxidation.

In addition, peeling Cu compound particles can become marine pollutants. There is no doubt that they will have different effects on marine organisms [33,34,35]. This study did not examine seafloor sediments, so the behavior of type C, which contains copper compounds, on the seafloor remains unclear. In the near future, we aim to conduct a survey of the seafloor.

## 3. Materials and Methods

### 3.1. Materials

The PP pellet sample was supplied by Prime Polymer Co., Ltd., Tokyo, Japan (product name: J-700GP). The weight-average molecular weight (Mw) and molecular weight distribution (Mw/Mn) were 2.9 × 10^5^ and 5.7. Potassium persulfate (K_2_S_2_O_8_) and cuprous oxide (Cu_2_O) were purchased from Wako Pure Chemical Industries (Osaka, Japan). Seawater was prepared with Gex artificial saltwater purchased from Amazon.co.jp (Tokyo, Japan. https://www.amazon.co.jp/gp/product/B09R9x4Y43/ref=ppx_yo_dt_b_asin_title_o02_s00?ie=UTF8&th=1, accessed on 22 November 2022).

### 3.2. Characterization

#### 3.2.1. Scanning Electron Microscope (SEM) with Energy-Dispersive X-ray Spectroscopy Analysis

The SEM/EDX analysis was carried out with a JSM-7500FAM (JEOL, Tokyo, Japan) at 5.0 kV. The working distance was about 3 × 4 mm. Samples were placed in a drying oven, maintained at 27 °C for 30 min and were sputter-coated with gold before SEM imaging.

#### 3.2.2. Fourier Transform Infrared (FT-IR) Analysis

The IR spectra 16 scans were measured with an FT-IR spectrometer (Jasco FT-IR 660 plus) at a resolution of 4 cm^−1^ over the full mid-IR range (400–4000 cm^−1^). The carbonyl index (CI) of PP was calculated as the band intensity ratio of carbonyl group (ca. 1720 cm^−1^)/scissoring CH_2_ group (ca. 1452 cm^−1^) [36,37,38].

#### 3.2.3. Gel Permeation Chromatography (GPC) Analysis

PP samples were dissolved in 5 mL of *o*-dichlorobenzene and were measured at a concentration of 0.5 mg/mL. Since small insoluble substances were observed in the sample solution, they were removed by heat filtration using a sintered filter with a pore diameter of 0.5 μm, and only the soluble portion was subjected to GPC measurement. The molecular weights were directly measured by HLC-8321GPC/HT GPC system (Tosoh Co., Ltd., Tokyo, Japan) at 140 °C and determined by a weight-average molecular weight in terms of PP.

### 3.3. Methods

#### 3.3.1. MP Sampling from the Sea

MP retrieving was carried out on 26 July 2021 using a training vessel T/V Kakuyo-maru (155 gross tonnage: Faculty of Fisheries, Nagasaki University) [39]. The bottom paint on the recovery vessel was silicon-based, and no copper-based paint was used for the purpose of this collection. The sampling vessel is made of polyvinyl chloride and its surface is not coated with paint. The 5 sampling stations in the East China Sea (between Nagasaki port and Island, Japan) are shown in Appendix A. Each sampling of 1 L seawater was carried out in one day. The samples denoted as “D” were collected using a Conductivity Temperature Depth profiler (CTD) system at ca. 50 m below the sea level, and the surface sampling (denoted as “B”) was carried out with a 3 L stainless bucket at 3 m below sea level.

#### 3.3.2. Filtration for SEM/EDX Observation of MP Retrieved from the Sea

Filtration was carried out with a polycarbonate membrane filter (Merck Isopore™ membrane, Sigma-Aldrich Japan Co., LLC., Tokyo, Japan) with 0.8 μm pore size without pretreatment.

#### 3.3.3. Preparation of PP Sample Containing Cuprous Oxide (Cu_2_O)

Since PP pellet particles varied in size, they were crushed and sieved through a 200 mesh (0.71 mm) sieve to equalize their size. A 2 g powdery sample of PP was put into a 50 mL glass vessel, and then a predetermined amount of Cu_2_O tiny particles (particle size = ca. 2 μm) was added to it. Both of them were uniformly melt-mixed and compress-molded into a film as a sample containing cuprous oxide.

### 3.4. Enhanced Degradation Method

The method used was a sulfate ion radical-induced degradation of PP samples containing Cu_2_O in seawater, which we named the “enhanced degradation method”. The PP samples containing Cu_2_O were compress-molded into thin films (30 × 30 × 0.060 mm) by compression molding at 180 °C under 10 MPa for 11 min. The degradation was performed in seawater using sulfate ion radical. The procedure was according to reports [15,25]. (1) Five pieces of each film were put into a 100 mL glass vessel containing 20 mL of seawater solution with 0.54 g K_2_S_2_O_8_ at ca. 65 °C for 12 h under stirring with a stirrer tip speed of ca. 100 rpm. (2) An equal amount of K_2_S_2_O_8_ seawater solution was added to compensate for the consumption of oxidant, and its degradation was carried out for 12 h under the same conditions. (3) The five pieces of the film were then transferred to a new 100 mL glass vessel containing a 20 mL of seawater solution with 0.54 g K_2_S_2_O_8_, and the degradations were restarted under the same conditions. The enhanced degradation method was carried out for a predetermined number of 15 days using (1) to (3) as one set (total 15 sets). The pH value of the solution was changed from 8.2 to 3 during each set.

## 4. Conclusions

Marine MPs and microdebris were retrieved from the five sampling stations between Nagasaki port and Goto Island and were classified into six types by elemental analysis using SEM/EDX. Three of these types, MP (A), Si-based (B), and Cu-based (C) paint particles, were found to be predominant in the sea. Type C contained cuprous oxide with high specific gravity and no depth dependence. Conversely, types A and B were mainly found on the sea surface and in the sea, respectively, with specific-gravity dependence. The type-A, i.e., MP, particles were very small, with 74% of them having a long diameter of 25 µm or less. The dominant size of type C was less than 7 μ m and contained 31–40 mol% of Cu content. Type C stayed at shallow depths up to 50 m below the sea surface for a long time. The long-stay behavior had consequences for degrading the MP with low specific gravity. To clarify the catalytic action of type C on the degradation, PP film samples containing cuprous oxide were prepared, and accelerated degradation behavior in seawater was investigated with the enhanced degradation method using the sulfuric acid radical initiator in seawater of various salinity concentrations. The IR spectrum of the degraded PP sample with cuprous oxide revealed that the copper soap compound was produced in seawater. The SEM/EDX analysis results indicated that the chemical reactions between Cl^−^ and cuprous oxide produced Cu^+^ ions. The higher-salinity concentrations affected not only Cu^+^ species but also ClOH production. The acceleration of PP degradation was caused by Cu^+^ functioning in a small amount. Cu^+^ adsorbed by ionic bonds as copper carboxylate and/or copper hydroperoxide was frequently released because of equilibrium and generated OH• according to the reaction with ClOH. The CI values obtained by the degradation of the samples in seawater were higher than those obtained in pure water up to the 6-day degradation time, regardless of the salinity concentration. Since the side reactions producing esters and carboxylic acids were affected by various factors such as pH values, the degree of progress of the side reactions after the 6-day degradation time differed depending on the salinity concentration. The number of polymer chain scissions was estimated from the 1/Mn values of each PP and PP sample containing 5 phr cuprous oxide in normal and three-times-salinity concentrations of seawater. Both samples containing cuprous oxide at normal and three-times-salinity concentrations had values approximately 30–50% higher than those without cuprous oxide after up to 9 days. Conversely, there was almost no difference in the value after 15 days among these samples, showing that the adsorbed copper ions were removed by repeated solution exchange. These results support the hypothesis that Cu^+^ species promote autoxidation.

## Figures and Tables

**Figure 1 molecules-29-01173-f001:**
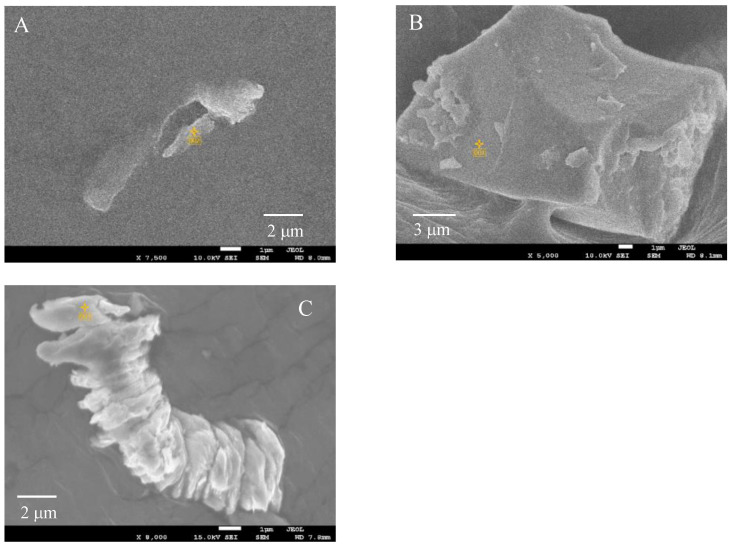
SEM photographs of samples retrieved by the sea: (**A**): MP. (**B**): Si-based paint particle. (**C**): Cu-based paint particle. The yellow symbols refer to the locations where EDX measurements were performed for sample assignment.

**Figure 2 molecules-29-01173-f002:**
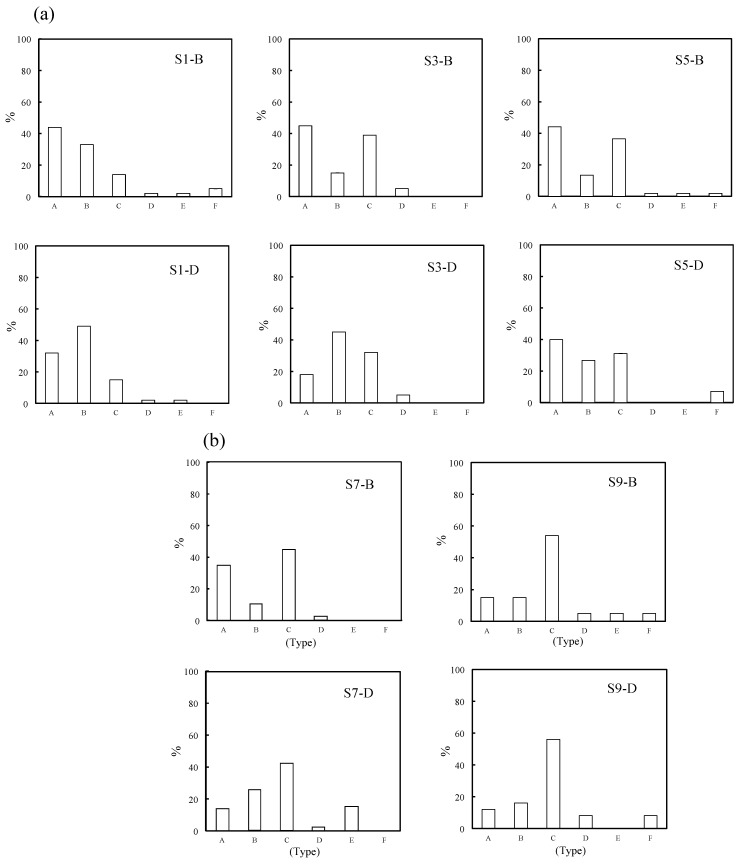
Percentage of microplastic and microdebris (**a**) at S1–S5 and (**b**) at S7 and S9 sampling stations, respectively. Type A: Microplastics. Type B: Si-based paint particles. Type C: Cu-based paint particles. Type D: Other paint particles. Type E: Al coating laminate film debris. Type F: Shell pieces.

**Figure 3 molecules-29-01173-f003:**
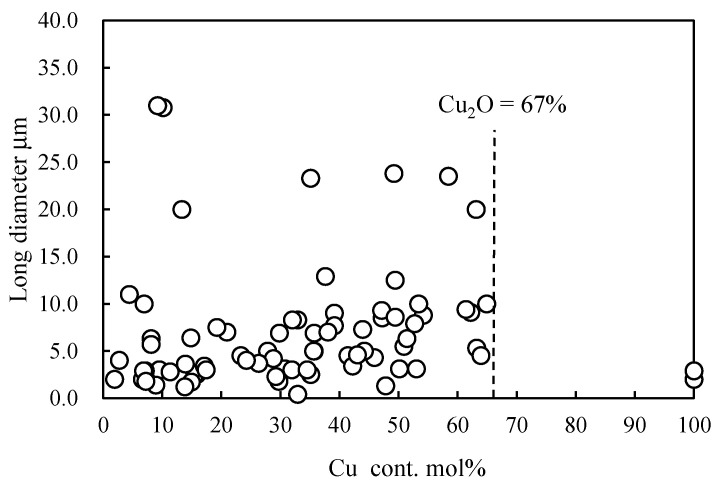
Relationship between Cu content ratio and long diameter of type-C (Cu paint particle) samples retrieved from the sea.

**Figure 4 molecules-29-01173-f004:**
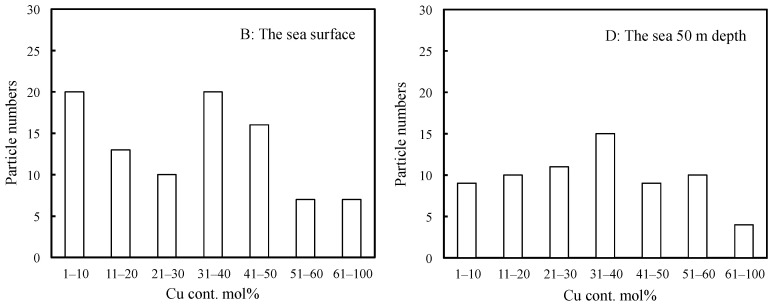
Particle numbers of type C and their Cu content ratio in the B and D spots at all sampling stations.

**Figure 5 molecules-29-01173-f005:**
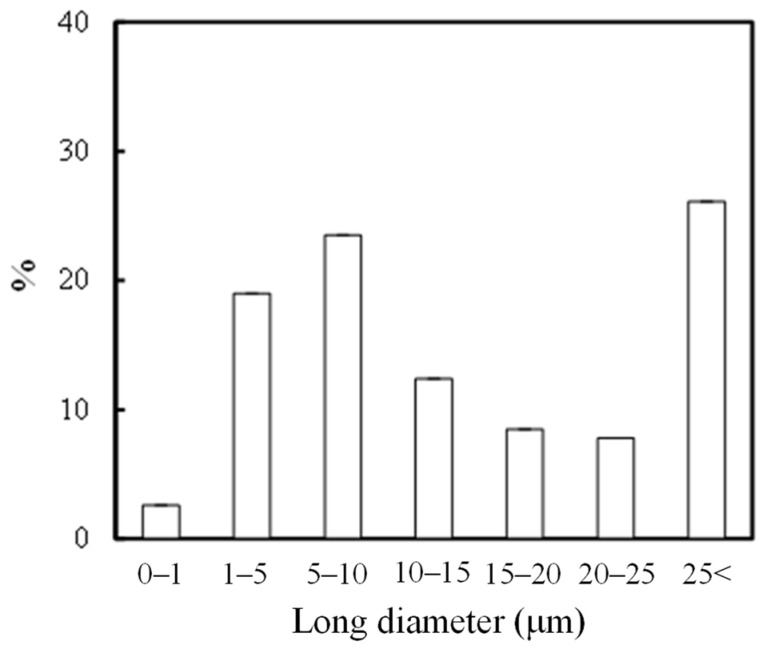
Percentage of long diameter of microplastic (Type A) samples retrieved from the sea.

**Figure 6 molecules-29-01173-f006:**
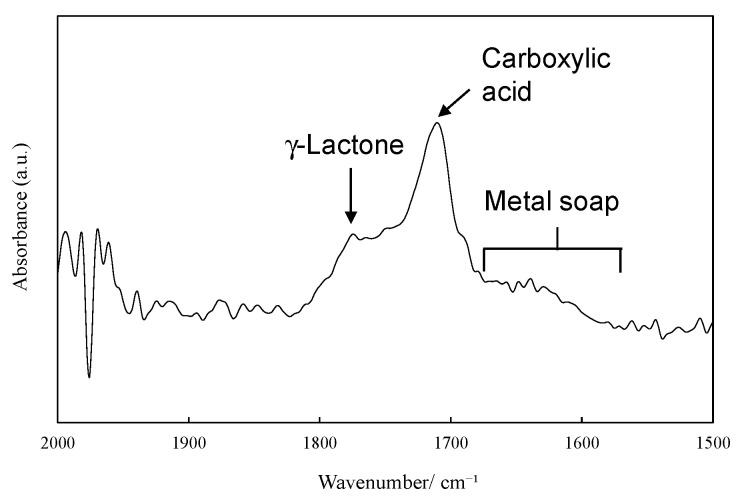
A reaction path of metal soap production via PP autoxidation and FT-IR spectrum of degraded PP containing 5 phr cuprous oxide using the enhanced degradation method in seawater for 3-days.

**Figure 7 molecules-29-01173-f007:**
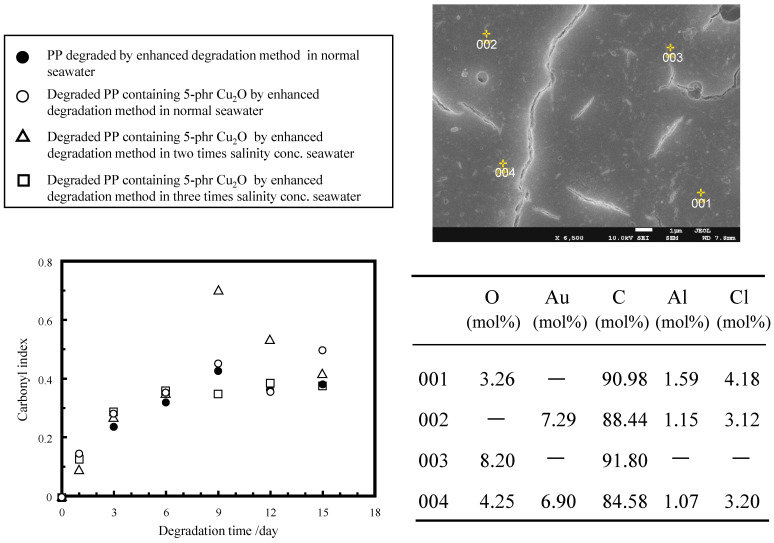
Degradation time dependences of FT-IR spectra and carbonyl index values of various PP samples in various salinity concentrations of seawater, and SEM/EDX analysis of degraded PP containing 5 phr Cu_2_O by the enhanced degradation method in the normal salinity concentration of seawater for 6-days.

**Figure 8 molecules-29-01173-f008:**
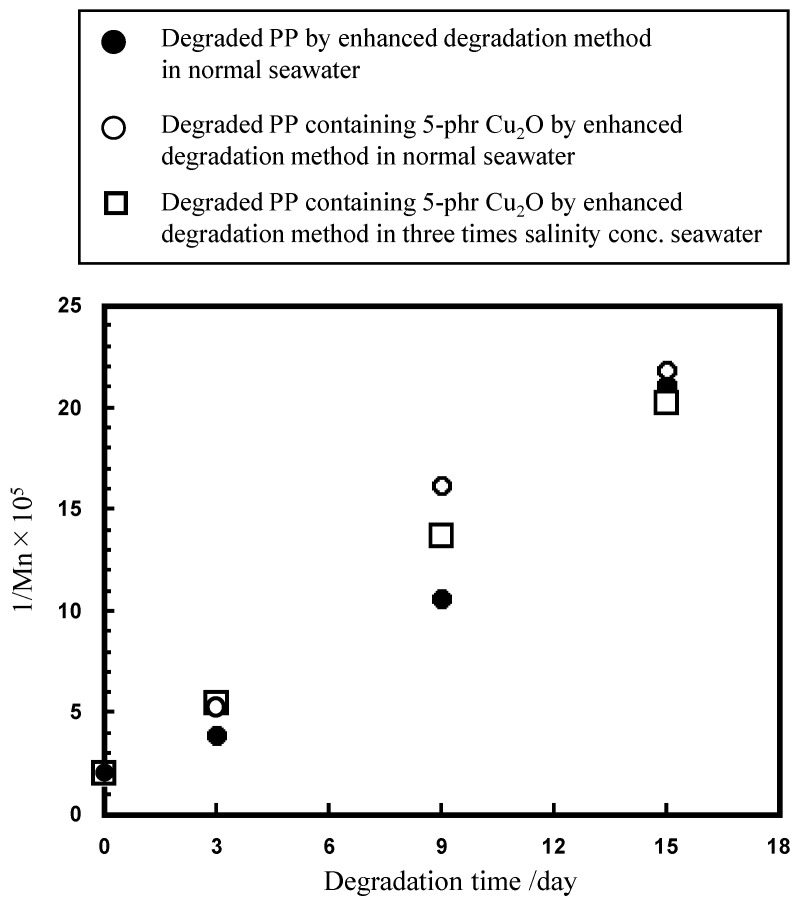
1/Mn × 10^5^ versus degradation time plots in various samples.

**Table 1 molecules-29-01173-t001:** Molecular weight information of various samples.

Samples	Deg. Days	Mn	Mw	Mz	Mw/Mn	Mz/Mw
PP original	0	50,120	286,262	953,225	5.7	3.3
Degraded PP in normal seawater	3	25,792	109,773	263,810	4.3	2.4
	9	9438	30,076	65,159	3.2	2.2
	15	4727	9950	18,526	2.1	1.9
Degraded PP containing 5 phr Cu_2_O in normal seawater	3	18,044	76,769	187,614	4.3	2.4
	9	6187	15,258	30,579	2.5	2.0
	15	4582	9589	17,767	2.1	1.9
Degraded PP containing 5 phr Cu_2_O in three times salinity conc. seawater	3	18,374	71,490	163,820	3.9	2.3
	9	7282	19,889	40,817	2.7	2.1
	15	4935	10,256	18,686	2.1	1.8

## Data Availability

Data are contained within the article.

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
