# Peer review of "Effect of Copper Antifouling Paint on Marine Degradation of Polypropylene: Uneven Distribution of Microdebris between Nagasaki Port and Goto Island, Japan"

_molecules, 2024, doi:10.3390/molecules29051173_

Round 1

Reviewer 1 Report

Comments and Suggestions for Authors

The method and writing of this manuscript are both well. I focus on two small issues. First, Discussion should discuss the impact of the material falling off, rather than just the process of falling off. There is no doubt that heavy metals and microplastics will have different effects on Marine organisms; second, authors should structure past manuscripts in this journal.

Author Response

Thank you very much for your kind suggestion! We added some sentences and the corresponding references at end of Discussion.

Reviewer 2 Report

Comments and Suggestions for Authors

The manuscript is a welcomed addition to the micro plastic related study, particularly the surface chemistry of the particles. However, the manuscript lack of international interest/broader interest and 6 sampling points is not clearly represented the area surrounding Nagasaki, thus the Title could be a misleading to the readers. 

The abstract need to be improved- starting the abstract with research design sounds too hurried, please add one-two introductory sentence with gaps.

Additionally, the  introduction also could be improved, especially to include the research gaps and knowledge in the surface chemistry of micro plastics. 

How 6 sampling points could represent a whole scenario in the marine waters, this need to be critically discussed. 

Line 197 and 251: the captions of the figures 6 and 7 seems partly-missing?

Table 8, please separate the table Mw and figure degradation rate. 

Method: need to clear- why 0.71 mm (710 um) PP was chosen? is this size similar to easily found in the marine water/paint dispersion? additionally this size also need to be discussed. 

Author Response

The manuscript is a welcomed addition to the micro plastic related study, particularly the surface chemistry of the particles. However, the manuscript lack of international interest/broader interest and 6 sampling points is not clearly represented the area surrounding Nagasaki, thus the Title could be a misleading to the readers.

Answer: Thank you very much for your kind suggestion! We revised Title as follows: Effect of Copper Antifouling Paint on Marine Degradation of Polypropylene: Uneven Distribution of Microdebris between Nagasaki port and Goto island, Japan.

The abstract need to be improved- starting the abstract with research design sounds too hurried, please add one-two introductory sentence with gaps.

Answer: We added the sentence “Microplastics (MP) are not only plastic products but also paint particles.“ in Abstract.

Additionally, the introduction also could be improved, especially to include the research gaps and knowledge in the surface chemistry of micro plastics.

Answer: We added the sentences in Introduction as follows : Very little research has been done on microplastics and their interactions with heavy metal paint fragments, i.e. the surface chemistry of microplastics.

How 6 sampling points could represent a whole scenario in the marine waters, this need to be critically discussed.

Answer: We added the sentences in Introduction as follows: Nagasaki Port is a port for small high-speed vessels (fishing boats), and the sampling station on the Goto side is located on the route of large vessels (ferries). The small vessels use silicon-based paint and the large vessels use copper-based paint, respectively. Therefore, these five sampling stations are expected to have different paint fragment distributions.

Line 197 and 251: the captions of the figures 6 and 7 seems partly-missing?

Answer: We revised these figure captions.

Table 8, please separate the table Mw and figure degradation rate.

Answer; We separated them.

Method: need to clear- why 0.71 mm (710 um) PP was chosen? is this size similar to easily found in the marine water/paint dispersion? additionally this size also need to be discussed. 

Answer: We added the sentences in Method as follows: Since PP pellet particles varied in size, they were crushed and sieved through a 200 mesh (0.71 mm) sieve to equalize their size.

The sieves of 0.71mm size has often been used for mixing polymers and pigments. This mixture is melt-mixed and has nothing to do with the size of the paint flakes.

Round 2

Reviewer 2 Report

Comments and Suggestions for Authors

Well done for the improvement, I only have one more suggestion:

1. Gel permeation chromatography (GPC) analysis - please add the specification for GPC column used here and HLC/GPC parameter/method. 

Author Response

Dear Reviewer 2,

Your comment: Gel permeation chromatography (GPC) analysis - please add the specification for GPC column used here and HLC/GPC parameter/method.

Our answer: We added them in GPC analysis.

Sincerely yours,

Hisayuki Nakatani